# Host Protease Activity on Bacterial Pathogens Promotes Complement and Antibiotic-Directed Killing

**DOI:** 10.3390/pathogens10111506

**Published:** 2021-11-18

**Authors:** Shaorong Chen, Dongmei Zhang, Alexandria-Jade Roberts, Hsueh-Chung Lu, Carolyn L. Cannon, Qing-Ming Qin, Paul de Figueiredo

**Affiliations:** 1Department of Microbial Pathogenesis and Immunology, Texas A&M Health Science Center, Bryan, TX 77807, USA; srchen@tamu.edu (S.C.); dongmei@tamu.edu (D.Z.); jamp1379@gmail.com (A.-J.R.); flu0302@gmail.com (H.-C.L.); carolyn.cannon@tamu.edu (C.L.C.); 2Department of Veterinary Pathobiology, Texas A&M University, College Station, TX 77843, USA

**Keywords:** host immune system, protease neutrophil elastase, bacterial pathogens, multidrug resistant bacteria, host complement system, antibiotics

## Abstract

Our understanding of how the host immune system thwarts bacterial evasive mechanisms remains incomplete. Here, we show that host protease neutrophil elastase acts on *Acinetobacter baumannii* and *Pseudomonas aeruginosa* to destroy factors that prevent serum-associated, complement-directed killing. The protease activity also enhances bacterial susceptibility to antibiotics in sera. These findings implicate a new paradigm where host protease activity on bacteria acts combinatorially with the host complement system and antibiotics to defeat bacterial pathogens.

## 1. Introduction

The human immune system deploys distinct innate immune mechanisms to thwart bacterial pathogens. These mechanisms include the host complement system and host proteases that are present at sites of bacterial infection. The complement system, a network of proteins in sera that are activated by microbial patterns, provides a first line of immune defense [1]. Activation results in the deposition of complement proteins on bacterial surfaces, thereby labeling bacteria for phagocytic uptake and subsequent killing. In addition, complement deposition on bacteria drives the formation of the membrane attack complex on bacterial surfaces, which kills Gram-negative bacteria via pore formation. Host proteases, released by immune cells, also contribute to bacterial killing by compromising the integrity of bacterial cell walls [2]. In addition, these proteases can destroy virulence factors and thereby thwart bacterial pathogenesis [3]. To survive within the human host, pathogens have evolved systems to circumvent, subvert, or evade these innate immune defense mechanisms [4,5,6,7,8,9,10]. However, the ways in which the host immune system can overcome these immune-evasive bacterial factors constitute a gap in our understanding. Here, we demonstrate that host protease activity on bacterial cell surfaces can destroy bacterial-associated complement inhibitory activities, thereby rendering resistant bacteria susceptible to complement-directed killing and sensitive to frontline antibiotics. Our studies featured the use of three clinically or agriculturally significant bacterial species: *Pseudomonas aeruginosa* (Pa), *Acinetobacter baumannii* (Ab), and *Brucella melitensis* (Bm). These species represent important bacterial pathogens that cause prevalent infectious diseases. Pa and Ab are opportunistic bacterial pathogens that constitute significant threats to civilian and warfighter personnel, as well as patients with underlying diseases, including cystic fibrosis [11,12,13]. Importantly, a significant proportion of clinical isolates of these pathogens display resistance to killing by normal human serum [14]. Moreover, we analyzed a vaccine strain of Bm, the world’s most prevalent bacterial zoonotic agent that displays complement resistance [15].

## 2. Results and Discussion

To test the hypothesis that protease activity on bacteria confers enhanced sensitivity to complement killing in human serum, we used a checkerboard strategy [16,17] to assess the effect of interaction between neutrophil elastase (NE) and pooled human complement serum (HS) on bacterial killing. Briefly, we co-incubated the above-mentioned bacterial strains at 1 × 10^3^ bacteria in the presence of various concentrations of HS (0–20%) and NE (0–0.3 U/mL) at 37 °C. We then determined bacterial growth by measuring the OD_600_ of the culture using a plate reader or CFUs (colony-forming units) on solid Luria–Bertani plates at 16 (Pa and Ab) or 72 (Bm) hr post-inoculation (h.p.i.) to assess the inhibitory or combinatorial activity of HS and NE on the survival or growth of the tested bacteria. Pa strain PAO1 displayed significantly reduced turbidity in 7.5% HS and poorly grew or even stopped growing when co-treated with 7.5% HS and 0.05 U/mL of NE (Appendix A). In 5% HS and 0.1 U/mL of NE, the strain displayed poor growth; moreover, the turbidity of the cultures was significantly reduced by 0.3 U/mL of NE (Figure 1A; Appendix A). Pa strain PA14 was weakly resistant to 2.5% HS and growth of this strain was strongly inhibited when treated with greater than 0.1 U/mL NE at this concentration of HS (Appendix A). To explore the hypothesis that NE targeting of protease-labile components on bacterial cell surfaces promoted complement directed killing, we performed similar experiments using PAO1 strains that harbored mutations in the *Ecotin*, *Wzz,* or *AprI* genes. *Ecotin* and *Wzz* contribute to complement resistance [18,19]. AprI is an inhibitor of AprA that protects PAO1 from complement killing. We found that deletion of *Ecotin* or *Wzz* significantly increased sensitivity of the mutants to 5% HS (Figure 1B,C), indicating that both *Ecotin* and *Wzz* genes were required for Pa resistance to complement-directed killing. In addition, the PAO1∆*AprI* mutant displayed enhanced resistance to complement killing and grew well in 5% HS, but was weakly inhibited in 10% HS (Figure 1D). Ab strain Ab5075, a highly virulent and multiple first-line antibiotic-resistant isolate of *A. baumannii* [20,21], displayed resistance to 5% HS but significantly increased sensitivity when treated with NE from 0.1 to 0.3 U/mL (Figure 1E). When treated with 0.3 U/mL NE, the strain displayed significantly reduced growth in 1.25% HS, strong growth inhibition in 2.5% HS, and extremely poor or even no growth in 5% HS (Figure 1E). Similar inhibitory or combinatorial activity of HS and NE on the growth of PAO1 (Figure 1F) or Ab5075 (Figure 1G) was observed when the approach of a CFU counting assay was used. The Bm vaccine strain Bm16MΔ*vjbR* displayed more resistance to complement killing than Ab5075 and PAO1. Bm16MΔ*vjbR* displayed growth in 15% HS in the presence of NE concentrations of ≤ 0.05 U/mL; however, NE treatment increased bacterial sensitivity to HS. When treated with more than 0.1 U/mL of NE, the growth of the Bm strain was significantly inhibited (Figure 1H). Interestingly, Bm16MΔ*vjbR* grew better in HS (<10%) than a non-HS-containing medium (Figure 1H). To verify that a heat-labile proteinaceous component of HS was mediating the observed growth-inhibition or killing activity, we measured bacterial survival in reaction mixtures that contained heat-treated HS. Briefly, HS was heat-inactivated at 55 °C for 0.5 h or 65 °C for 1 h and then incubated with PAO1 or Ab5075. Under these conditions, no inhibition of bacterial growth was observed, and the combinatorial effect observed with non-heat-killed HS was eliminated (Figure 1I–J; Appendix A). Interestingly, NE simultaneous coincubation with HS resulted in a better bacterial growth inhibition than a subsequent addition of HS (Figure 1K–L), suggesting that a longer period of NE and HS coincubation yields a better combinatorial effect in HS (2.5–5%). The additional Ca^2+^ or Mg^2+^ in the media significantly inhibited PAO1, but not Ab5075, growth only in the presence of HS (>5%) (Appendix A).

Taken together, the data support the hypothesis that the combinatorial interactions between protease activities and complement-directed growth inhibition or killing promote the destruction of Gram-negative bacterial pathogens. We were intrigued whether these findings could be extended by determining whether protease activity on multi-drug-resistant bacteria could enhance sensitivity to front-line antibiotics. Toward this end, we measured the growth of bacteria in the presence of ampicillin, tobramycin or carbenicillin. Ab5075 was resistant to ampicillin (Appendix A), weakly resistant to 25 µg/mL tobramycin (Figure 2A); however, both PAO1 and PA14 displayed significant sensitivity to tobramycin at this concentration (Figure 2B–C). NE (0.1 to 0.3 U/mL)-treated Ab5075 was susceptible to 25 µg/mL tobramycin in 2.5% HS and this treatment displayed a combinatorial effect on bacterial killing (Figure 2D); NE-treated PAO1 and PA14 stopped growing in 12.5 µg/mL tobramycin (Figure 2E–F). PAO1 and PA14 were weakly resistant to carbenicillin (<16 µg/mL) and the presence of NE significantly increased the sensitivity of these bacteria to carbenicillin (Figure 2G). Collectively, these data demonstrated a combinatorial interaction between protease activity on bacteria and antibiotic treatment in driving the killing of bacterial pathogens, providing a new avenue for bacterial disease management.

To test the potentiality of our findings in complementation with the existing antibiotic treatments, we used the fluorescence microscopy assay to evaluate the cytotoxicity of human NE to human THP-1 cells, a leukemia monocytic cell line that has been extensively used to study monocyte/macrophage functions, mechanisms, signaling pathways, and nutrient and drug transport [22]. Our results indicated that human NE did not induce any cytotoxicity to THP-1 cells, compared to the untreated control (Appendix A). The data suggest that human NE is not toxic to human cells.

In this report, we demonstrated that host protease activity on bacteria acts in a combinatorial fashion with the host complement system and antibiotics to defeat bacterial pathogens. The antibiotics used in this work included front-line antibiotics used in the clinic. We also showed that human HS and NE are not toxic to human cells. Therefore, it may be feasible to develop host protease treatments, together with antibiotics and/or HS, to treat infections. Of course, native host protease activity may contribute to fighting multidrug resistant bacterial pathogens, such as *A. baumannii* strains. However, additional work needs to be done to define how best to exploit host protease activity to defeat infections in the clinic. Future work will focus on these aspects.

## 3. Materials and Methods

### 3.1. Bacterial Strains and Culture

Bacterial strains used in this work included *A. baumannii* 5075 (Ab5075); *P. aeruginosa* strains PAO1, PA-14, PAO1Δ*ecotin*, PAO1Δ*Wzz*, and PAO1Δ*AprI;* and *B. melitensis* vaccine candidate strain Bm16MΔ*vjbR.* The origin and features of the strains are listed in Appendix A. The bacterial strains were stored in a – 80 °C freezer and were short-term maintained on Luria–Bertani (for Pa and Ab strains) or on tryptic soy agar (TSA, Difco™) (for *Brucella* strain) [23] plates.

### 3.2. Checkerboard Test

Checkerboard tests were performed as described in the literature with minor modifications [16,17]. Briefly, a single colony of each tested bacterial strain was inoculated in 5 mL Mueller–Hinton Broth (MH, Sigma-Aldrich, 70192-500G) or Cation-adjusted Mueller–Hinton Broth 2 (MH2, Sigma-Aldrich, 90922-500G) and incubated overnight at 37 °C in a shaker at 225 rpm. The overnight cultures (100 μL/each strain) were transferred to 5 mL fresh MH or MH2 broth and incubated at 37 °C in a shaker at 225 rpm until the culture reached a visible turbidity that was equal to or greater than the turbidity of a McFarland Standard 0.5. OD_600_ of each culture was determined and adjusted to 1 × 10^4^ bacteria/mL. The tested bacteria (100 μL) were then mixed with 100 μL MH or MH2 containing a gradually increasing concentration of pooled human complement serum (HS, Innovative Research, ICSER100ML; 0–20%), human neutrophil elastase (NE, EMD Millipore, 324681-100UG; 0–0.3 U/mL) or antibiotic tobramycin (TCI Chemicals T2503-5G; 0–100 μg/mL), or carbenicillin (GOLDBIO, C-103-50; 0–32 μg/mL) in 96-well plates. The plates were then incubated in a growth chamber at 37 °C for 16 hr (72 hr for Bm16MΔ*vjbR*). The growth of the tested bacteria was visualized for the turbidity, and OD_600_ values of each well were determined by a Cytation 5 imaging reader (Biotek, Inc., Winooski, VT, USA). The growth of the tested bacteria in 96-well plates was also photographically documented. In each experiment, the tested bacterial strain treated with the lowest concentration of HS, NE, or/and antibiotic tobramycin or carbenicillin served as a control. Growth inhibition rate (GIR, %) was calculated as the following formula: GIR % = [(Control OD_600_–treatment OD_600_)/Control OD_600_] × 100%. The colors were designated based on the GIR values; “[”or“]” and “(”or“)” indicate inclusion and exclusion, respectively.

### 3.3. CFU Assay

A CFU assay was performed as the method previously described [23] with a minor modification. Briefly, the tested bacterial cells were co-incubated with different concentrations of NE and/or HS as mentioned above in 96-well plates in a growth chamber at 37 °C. At 16 h post incubation, bacterial cells from each well were subjected to a series of 10× dilutions (performed in a biosafety cabinet), and the serial dilutions (10 μL) were dropped on solid Luria–Bertani plates. The plates were gently turned around to allow the bacterial droplets to spread over and dry more quickly. The plates were then incubated in a growth chamber at 37 °C. The colonies on the plates were counted and documented at 16 h post incubation.

### 3.4. Host Cell Cytotoxicity Assay

A human NE host cell toxicity assay was performed as the method described previously [24] with minor modifications. Briefly, human THP-1 cells (3 × 10^5^ cells/mL, 200 μL) with (0.3 U/mL) or without NE were seeded in the wells of a 96-well plate, and the plate was then incubated at 37 °C, 5.0% CO_2_ for 24 h. The cells were then stained with propidium iodide (PI, 0.5 µg/mL) and Hoechst (5.0 µg/mL) for 1 h in the incubator. The stained cells were imaged using an immunofluorescence microscopy system (Nikon Eclipse Ti2, USA). The dead cells with PI and Hoechst were enumerated from the collected images using Ti2 software in the microscopy system. The rate of viable cells was calculated as the following: viable cell rate (%) = (total cells-dead cells)/total cells. For each treatment in each experiment, a total of ~14,000 cells were counted.

### 3.5. Quantification and Statistical Analysis

All the quantitative data represented the means ± standard error of measurements (SEM) from at least three independent experiments. The significance of the data was assessed using the Student’s *t* test (for two experimental groups) or a one-way ANOVA test to evaluate the statistical differences of multiple comparisons of the data sets.

## 4. Conclusions

Our findings supported that host protease NE acted on bacteria to destroy factors that prevented serum-associated, complement-directed growth-inhibition or killing, and that host protease activity enhanced bacterial susceptibility to antibiotics in sera. These findings implicated a new paradigm where host protease activity on bacteria acted combinatorically with the host complement system and antibiotics to defeat bacterial pathogens.

## Figures and Tables

**Figure 1 pathogens-10-01506-f001:**
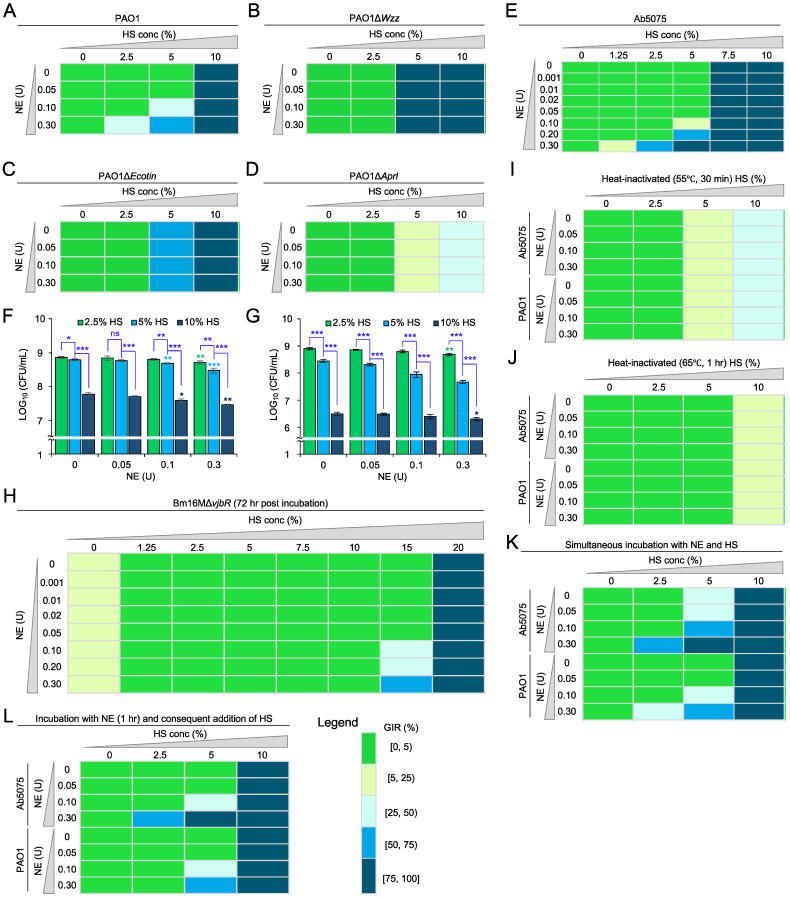
Combinatorial effect of neutrophil elastase (NE) and human serum on bacterial growth. The tested bacteria (1 × 10^4^ cells/mL, 100 μL) and 100 μL of gradually increasing concentrations of pooled Human Complement Serum (HS) were added into and mixed in each well of 96-well plates. The plates were then incubated at 37 °C and at 16 (for *P. aeruginosa* and *A. baumannii*) or 72 (for *B. melitensis* Bm16MΔ*vjbR*) hr post incubation, the growth of the treated bacteria was measured using a plate reader or via counting of colony-forming units (CFUs) on solid Luria–Bertani plates. The bacterial growth was mainly measured via a plate reader, unless otherwise indicated. (**A**–**D**) Combinatorial effect of protease and serum on bacterial pathogens *P. aeruginosa* PAO1 (**A**) and its derived mutants PAO1∆*Wzz* (**B**), PAO1∆*Ecotin* (**C**), and PAO1∆*AprI* (**D**). (**E**) NE promotes *A. baumannii* Ab5075 growth inhibition in the presence of HS. (**F**,**G**) NE promotes PAO1 (**F**) or Ab5075 (**G**) growth inhibition in the presence of HS. The growth of the treated bacteria was measured using a CFU counting assay. Data represent the means ± standard error of measurements (SEM) from three independent experiments; ns: no significance; *, **, and ***: significance at *p* < 0.05, 0.01, and 0.001, respectively. Green, light blue, and dark blue asterisks: compared to the growth of bacterial cells treated with 0 U NE and with 2.5%, 5%, and 10% HS, respectively. (**H**) NE promotes Bm16MΔ*vjbR* growth inhibition in the presence of HS. (**I**,**J**) Bacterial growth-inhibition effect of NE decreased in the presence of heat-treated serum at 55 °C for 30 min (**I**) or at 65 °C for 1 h (**J**). (**K**,**L**) Addition of HS in NE at the same time (**K**) or at 1 h post incubation of NE and bacteria (**L**) varied the combinatorial effect. Growth inhibition rate (GIR, %) = [(Control OD_600_–treatment OD_600_)/Control OD_600_] × 100%. “[”or“]” and “(”or“)” indicate inclusion and exclusion, respectively; conc: concentration.

**Figure 2 pathogens-10-01506-f002:**
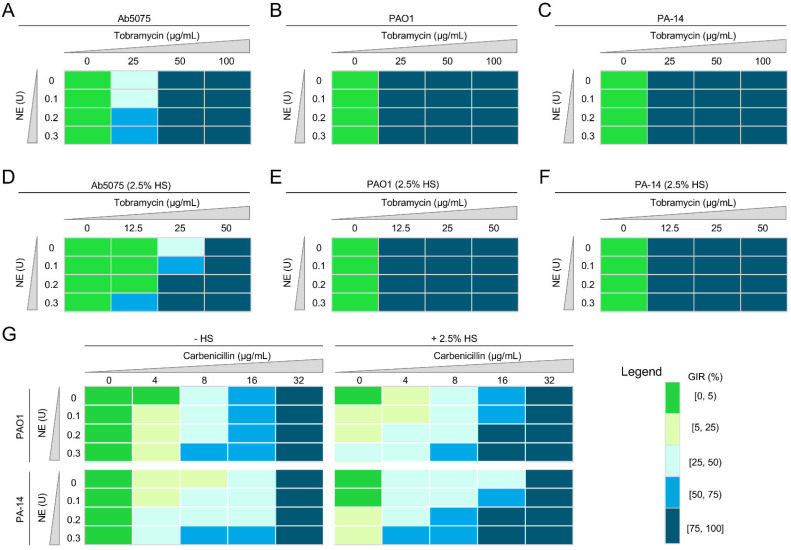
Combinatorial effect of NE and tobramycin, or carbenicillin, and/or HS on bacterial killing. (**A**–**C**) NE promotes *A. baumannii* Ab5075 (**A**), *P. aeruginosa* PAO1 (**B**), or PA-14 (**C**) killing in the presence of tobramycin. (**D**–**F**) Combinatorial effect of NE and tobramycin on *A. baumannii* Ab5075 (**D**), but not on PAO1 (**E**) or PA-14 (**F**), killing in the presence of 2.5% HS. (**G**) Combinatorial effect of NE and carbenicillin on PAO1 and PA-14 killing in the presence of 2.5% HS.

## Data Availability

This study did not generate/analyze data sets or code.

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
