# Peer review of "Host Protease Activity on Bacterial Pathogens Promotes Complement and Antibiotic-Directed Killing"

_pathogens, 2021, doi:10.3390/pathogens10111506_

Round 1

Reviewer 1 Report

Revisions made in response to initial review are adequate.

Reviewer 2 Report

The hypothesis progressed herein that the host protease activity on bacteria acts combinatorial fashion with the host complement system and antibiotics to defeat bacterial pathogens, is supported by properly designed and executed scientifically sound experiments.

Though it’s proper to clearly state the significance of these findings in such a way that, 1) how these findings would complement the existing antibiotic treatments, particularly for the antibiotic-resistance bacterial infections – which is a currently unmet medical need, 2) Is it feasible to develop host protease treatment as a novel antibacterial therapeutic option, 3) and its limitations.

It’s proper to clearly distinctively state the control experiments in the methods, and the statistical significance of the treatment concentration to that of control in the result discussion.

It’s also recommended to show human cell toxicity data particularly towards HNE for the tested concentrations.         

Author Response

This manuscript is a resubmission of an earlier submission. The following is a list of the peer review reports and author responses from that submission.

Round 1

Reviewer 1 Report

This paper by Shaorong Chen and colleagues at Texas A&M investigates the role of neutrophil elastase and human serum (as a source of complement) in the growth inhibition of three GNRs: P. aeruginosa P01, Acinetobacter baumannii, and Brucella melitensis. They use a checkerboard dilution method in 96 well microtiter plates and read OD after 16 hours, or 72 hours for B. melitensis. The results are displayed as percent reduction compared to the OD of bacteria grow in Mueller Hinton Broth only. They also did colony counts for some of the combinations.

The writing and the illustrations are clear, and they included several controls such as antibiotic plus elastase, heat inactivated human serum, and mutants of P. aeruginosa that have short chains of LPS. They found greater growth inhibition when elastase was added to serum, but not to heated serum, and a slight lowering of the MIC of Acinetobactor when elastase was added to tobramycin sensitivity, that was abolished by adding 2.5% serum. The P. aeruginosa with LPS mutations were more susceptible to human serum, which is to be expected since C3 binds to the OH groups on LPS and the further they are from the inner membrane the less effect they have on viability of bacteria.

               I found several problems with this paper. The first is that from their colony counts there was no bacterial killing, only inhibition of growth. They inoculated 104 CFU and without treatment after 16 hours had ~ 8X108. With treatment there were about 107, which indicates growth inhibition, not killing. Plotting bacterial growth on a linear rather than a log scale can be very misleading. Complement usually kills bacteria in minutes, not hours so i find it hard to understand why they incubated the cultures for 16 hours, potentially allowing time for surviving bacteria to multiply. The second is that for some reason they used Mueller Hinton broth for this assay, but I don’t think there is enough Ca++ or Mg++ in MH broth to support optimal complement activity, and at low serum concentrations those ions will be diluted out. So there appears to be a biological effect of elastase and human serum on the bacteria, but not the effect that the authors claim. Overall, their result with the LPS mutants suggest the effect is related to LPS so what role does a protease play?

               There is no explanation offered for the inclusion of B. melitensis ΔvjbR in this paper. ΔvjbR is virulence factor as it is a transcriptional regulator involved in quorum sensing. What activity of this molecule is being targeted?

The other issue is the commercial elastase enzyme. I have no experience with this product, but I have had problems with collagenases that were mixtures of enzymes. The description in the online catalogue says it is a “major band” on the gel, which suggests there are other proteins that may be enzymes.

Reviewer 2 Report

Supplementary file 1 could not be accessed. An “error 404 – file not found” message is at that URL. Supplementary files could not be reviewed.

Line 118: At this time, only data with tobramycin could be reviewed. Why was tobramycin selected, and only that antibiotic shown in the results?

Line 122: I do not agree that there is any combinatorial effect of NE with tobramycin on PA01 or PA-14 based on the data shown. Susceptibility to tobramycin is the same regardless of the concentration of NE used. The problem is that the effect of 12.5 μg/ml of tobramycin is not shown in the absence of HS, only 25 and 50 μg/ml. In order to make this claim the comparative results with 12.5 μg/ml in the absence of HS should be shown. Would other antibiotics have had a more differential effect? This should also be corrected in the legend to Fig. 2, which states there is a combinatorial effect for PA01 and PA-14, but the comparison with equal concentrations of tobramycin are not shown.

Line 128: Remove “(C-D)” from beginning of the sentence.

Lines 144-145: Why were the bacteria treated with NE and HS twice? This is not described in the results section. From the way the text is written, it appears that the bacteria were pre-treated with NE, then incubated with HS. These bacteria were then tested again with various concentrations of NE and HS together. If this is not correct this section needs to be re-written. If correct, the bacteria have already been compromised by treating with NE and HS, so why are these bacteria treated again, only with the agents together?

Line 122: The legend of figure 2 is in error. NE-treated PA01 and PA-14 are shown in Fig. 2E and 2F, not 2D.

Minor comments:

Line 50: (NE) should be used the first time after “neutrophil elastase”.

Line 159: Were the bacteria dropped on the plate then spread over the plate? If not , then how were individual colonies counted? If so, this should be stated.